# Safety and Efficacy of Neoadjuvant Chemoimmunotherapy in Gastric Cancer Patients with a PD-L1 Positive Status: A Case Report

Alexandra V. Avgustinovich [1], Olga V. Bakina [2,3], Sergey G. Afanas'ev [1], Liudmila V. Spirina [1,3,*] and Alexander M. Volkov [1]

1     Cancer Research Institute, Tomsk National Research Medical Center, Russian Academy of Sciences, 5 Kooperativny Street, Tomsk 634050, Russia; aov862@yandex.ru (A.V.A.); afanasievsg@oncology.tomsk.ru (S.G.A.); dok75-75@mail.ru (A.M.V.)

2     Institute of Strength Physics and Materials Science of the Siberian Branch of the Russian Academy of Sciences, 2/4 pr. Akademicheskii, Tomsk 634055, Russia; ovbakina@ispms.ru

3     Medico-Biological Faculty, Division of Biochemistry and Molcecular Biology with Clinical Laboratory Diagnostics Course, Siberian State Medical University, 2, Moskovsky Trakt, Tomsk 634050, Russia

*     Correspondence: spirinalvl@mail.ru; Tel.: +7-9609758577

**Abstract:** Introduction: The landscape of gastric cancer treatment has changed owing to the widespread use of immune checkpoint inhibitors. Autophagy, involved in regulating the immune system, is a potential trigger of immunity in tumors. This study aims to find molecular-based evidence for the effectiveness of FLOT chemotherapy with immune checkpoint inhibitors in gastric cancer patients. Materials and Methods: Three patients with advanced gastric cancer received FLOT neoadjuvant chemotherapy with immunotherapy and surgery. IHC was used to determine the PD-L1 status. Real-time PCR was used to analyze expression patterns of transcriptional growth factors, AKT/mTOR signaling components, PD-1, PD-L1, PD-L2 and LC3B. The LC3B content was measured via Western blotting analysis. Results: The combination of FLOT neoadjuvant chemotherapy and immunotherapy was found to be efficient in patients with a PD-L1-positive status. Gastric tumors with a PD-L1-positive status exhibited autophagy activation and decreased PD-1 expression. Conclusions: FLOT chemotherapy combined with immune checkpoint inhibitors showed high efficacy in gastric cancer patients with a positive PD-L1 status. Autophagy was involved in activating the tumor immunity. Further research is needed to clarify the mechanism of effective anticancer treatment.

**Keywords:** gastric cancer; neoadjuvant chemotherapy; immunotherapy

## 1. Introduction

Gastric cancer (GC) is one of the most common malignant neoplasms [1,2], ranking sixth in terms of incidence (5.7%) and second (9.3%) in terms of mortality in Russia. The early diagnosis rate for gastric cancer remains low. The percentage of Stage II and III cancers ranges from 39.9% to 44.9% [3,4]. Surgery is currently the gold standard for gastric cancer treatment, supplemented by neoadjuvant or adjuvant chemoradiotherapy [3].

In recent years, a large number of studies have focused on searching for additional prognostic and predictive molecular markers [5,6]. The Cancer Genome Atlas (TCGA) pilot project, initiated in 2006, has investigated more than twenty types of cancers with poor prognosis. Based on the Cancer Genome Atlas Research Network, gastric cancers can be classified into several molecular subtypes:

(1)     Epstein–Barr virus positive tumors with PIK3CA mutations, DNA hypermethylation and JAK2, CD274 (also known as PD-L1) and PDCD1LG2 amplification (also known as PD-L2) (9%);

(2)     Microsatellite unstable tumors with mutations in oncogenes (22%);

(3)   Genomically stable tumors, found in diffuse gastric cancers with RhoA mutations or fusion (20%);

(4)   Chromosomal instable tumors with tyrosine kinases mutations (50%).

Despite the presence of molecular subtypes, there is a lack of effective molecular markers in ordinary clinical practice to predict the response to anticancer therapy [6]. Gastric cancers have three predictive markers (Her2neu (receptor of epidermal growth factor), PD-L (programmed cell death ligand), and MSI (microsatellite instability)) for individualized treatment. They are mostly prescribed for metastatic cancers [5]. Her2neu status has already become routine in clinical practice [6,7]. The use of immune checkpoint inhibitors in gastric cancer treatment shows promise [8]. The incidence of gastric cancer with MSI-High varies from 10 to 22% [9,10].

Patients with non-metastatic gastric cancer and MSI-High had higher overall survival compared to patients with microsatellite stability (MSS) cancers [10]. However, the effectiveness of targeted drugs in therapy for resectable gastric cancers is still unclear. Anti-PD1 drugs (nivolumab and pembrolizumab) as well as anti-Her2neu drugs (trastuzumab) are prescribed for treating metastatic gastric cancers [2]. Furthermore, the investigation into the biological features that determine the response to anticancer therapy is ongoing [6].

In this study, we present clinical cases of patients with resectable gastric cancer who underwent FLOT chemotherapy with immunotherapy. Our goal was to identify the biological features of effective anticancer therapy by taking a molecular-based approach. This study aims to find molecular-based evidence for the effectiveness of FLOT chemotherapy with immune checkpoint inhibitors in gastric cancer patients.

## 2. Materials and Methods

### 2.1. Clinical Characteristics of Patients

Three patients with gastric cancers were enrolled in our study. They underwent FLOT therapy with immune checkpoint inhibitors and achieved a complete pathological response (Table 1). The median age was 58.0 years (ranging from 38 to 63 years). We assessed the effectiveness of neoadjuvant FLOT chemotherapy with immunotherapy (pembrolumab 400 mg over 6 weeks) based on the RECIST 1.1 criteria (complete or partial response, stable disease, progressive disease). Two patients showed a partial response, while one patient showed stable disease.

**Table 1.** Clinical characteristics of patients included in the study.

| Age | Gender | Localization of the Tumor | PD-L1 (CPS) | The Volume of Surgical Intervention | TNM |
|---|---|---|---|---|---|
| 58 | male | body | 10 | radical distal gastrectomy | T3N0M0 |
| 38 | male | body and pyloric antrum | 25 | combined gastrectomy | T3N0M0 |
| 63 | female | body | 20 | radical total gastrectomy | T3N0M0 |

We evaluated chemotherapy tolerance using the NCIC common toxicity criteria grading system. Radical surgery was performed 4–8 weeks after completing the chemotherapy. The extent of surgical intervention depended on the tumor location. The frequency and nature of postoperative complications were assessed using the Clavien–Dindo scale. Nausea was observed in all three cases (100%). Two patients (66.6%) experienced Grade 1–2 hematological toxicity. These toxic reactions did not require a reduction in the initial drug dosage.

This work was approved by the Local Committee of Medical Ethics, Cancer Research Institute, Tomsk National Research Medical Center of the Russian Academy of Sciences, on the basis of Minute No. 5, dated 24 April 2019.

The Mandard score was used to assess tumor regression after neoadjuvant chemotherapy. Post-operative biopsy samples of normal gastric and tumor tissues were used for the investigation. Samples were reviewed separately by two independent pathologists. The PD-L1 status was determined using the SP263 test on the BenchMark ULTRA platform (Ventana, Roche, Indianapolis, IN, USA). The PD-L1 status was positive with CPS > 10 (Figure 1).

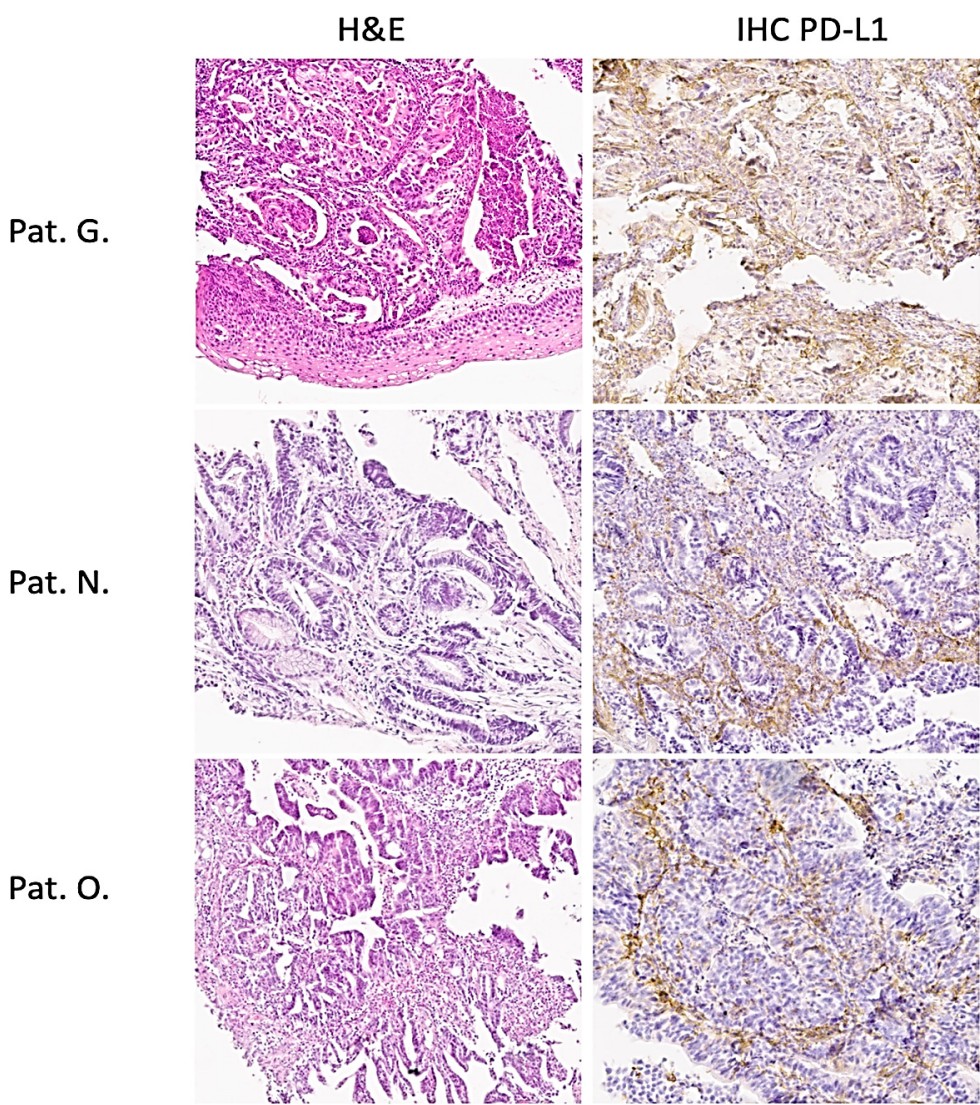

**Figure 1.** Histology and PD-L1 status in gastric cancer patients. H&E and IHC. Magnification 200× (Appendix A).

## 2.2. Molecular Characteristics of Tumors

### 2.2.1. RNA Extraction

Biopsy tissues were frozen and stored at t = 80 °C. The tumor samples were incubated in RNAlater solution (Ambion, Foster City, CA, USA) for 24 h at +4 °C and then stored at −80 °C. Total RNA was extracted using the RNeasy Mini Kit (Qiagen, Hilden, Germany).

PCR was performed in 25 μL reaction volumes containing 12.5 μL of BioMaster HS-qPCR SYBR Blue (2×) (Biolabmix, Novosibirsk, Russia) and 300 nM of each primer (Table 2).

**Table 2.** Primers sequences.

| Gene | Primers |
| --- | --- |
| *LC3B* | F 5′-CCCAAACCGCAGACACAT-3′, R 5′-ATCCCACCAGCCAGCAC-3′; |
| *mTOR* | F 5′-CCAAAGGCAACAAGCGAT-3′, R 5′-TTCACCAAACCGTCTCCAA-3′; |
| *AMPK* | F 5′-AAGATGTCCATTGGATGCACT-3′, R 5′-TGAGGTGTTGAGGAACCAGAT-3′; |
| *CAIX* | F 5′-GTTGCTGTCTCGCTTGGAA-3′, R 5′-CAGGGTGTCAGAGAGGGTGT-3′; |
| *HIF-1* | F 5′-CAAGAACCTACTGCTAATGCCA-3′, R 5′-TTTGGTGAGGCTGTCCGA-3′; |
| *EPAS1* | F 5′-TGGAGTATGAAGAGCAAGCCT-3′, R 5′-GGGAACCTGCTCTTGCTGT-3′; |
| *NFKB1* | F 5′-CGTGTAAACCAAAGCCCTAAA-3′, R 5′-AACCAAGAAAGGAAGCCAAGT-3′; |
| *RELA* | F 5′-GGAGCACAGATACCACCAAGA-3′, R 5′-GGGTTGTTGTTGGTCTGGAT-3′; |
| *VEGFA* | F 5′-AGGGCAGAATCATCACGAA-3′, R 5′-TCTTGCTCTATCTTTCTTTGGTCT-3′; |
| *KDR:* | F 5′-AACACAGCAGGAATCAGTCA-3′, R 5′-GTGGTGTCTGTGTCATCGGA-3′; |
| *4-BP1* | F 5′-CAGCCCTTTCTCCCTCACT-3′, R 5′-TTCCCAAGCACATCAACCT-3′; |
| *AKT1* | F 5′-CGAGGACGCCAAGGAGA-3′, R 5′-GTCATCTTGGTCAGGTGGTGT-3′; |
| *C-RAF* | F 5′-TGGTGTGTCCTGCTCCCT-3′, R 5′-ACTGCCTGCTACCTTACTTCCT-3′; |
| *GSK3b* | F 5′-AGACAAGGACGGCAGCAA-3′, R 5′-TGGAGTAGAAGAAATAACGCAAT-3′; |
| *70S kinase alpha* | F 5′-CAGCACAGCAAATCCTCAGA-3′, R 5′-ACACATCTCCCTCTCCACCTT-3′; |
| *PDK1:* | F 5′-TCACCAGGACAGCCAATACA-3′, R 5′-CTCCTCGGTCACTCATCTTCA-3′; |
| *VHL* | F 5′-GGCAGGCGAATCTCTTGA-3′, R 5′-CTATTTCCTTTACTCAGCACCATT-3′; |
| *PD-L2* | F 5′-GTTCCACATACCTCAAGTCCAA-3′, R 5′-ATAGCACTGTTCACTTCCCTCTT-3′; |
| *PD-L1* | F 5′-AGGGAGAATGATGGATGTGAA-3′, R 5′-ATCATTCACAACCACACTCACAT-3′; |
| *PD-1-1* | F 5′-CTGGGCGGTGCTACAACT-3′, R 5′-CTTCTGCCCTTCTCTCTGTCA-3′; |
| GAPDH | F 5′-GGAAGTCAGGTGGAGCGA-3′, R 5′-GCAACAATATCCACTTTACCAGA-3′. |

To activate the Hot Start DNA polymerase and denature DNA, a pre-incubation at 95 °C for 10 min was performed. This was followed by 45 amplification cycles consisting of denaturation at 95 °C for 10 s and annealing at 60 °C for 20 s (iCycler iQ™, BioRad, Hercules, CA, USA).

The fold changes were estimated using the ΔΔCt method, where the total ΔΔCt represented the fold change in the gene level in cancerous tissue compared to normal tissue. The ratio of specific mRNA/GADPH (GADPH as a control) amplification was then calculated.

### 2.2.2. Determination of LC3B Content

To determine the LC3B content, electrophoresis SDS-PAGE (Laemmli) was used. The protein was transferred to a PVDF membrane with a pore size of 0.2 μm (Uber, GE Healthcare, Solingen, Germany) at either at 150 mA or 100 V for 1 h using a Bio-Rad Mini Trans-Blot electrophoresis cell. The membrane was incubated overnight at 4 °C with a 1:2500 dilution of monoclonal mouse anti-human LC3B (Affinity Biosciences, Cincinnati, OH, USA).

PVDF samples were incubated using the Amersham ECL Western blotting detection analysis system (Amersham, Chicago, IL, USA). The results were standardized using beta-actin expression in a sample and expressed as percentages relative to the protein content in non-transformed tissues. The protein level in normal gastric tissue was considered as 100%.

### 2.2.3. Immunohistochemical Staining of PD-L1

The formalin-fixed and paraffin-embedded (FFPE) primary tumor samples were used for IHC staining. The monoclonal mouse anti-human PD-L1 antibody (Clone 22C3, Dako, Glostrup, Denmark, 1:50) was used as the primary antibody. The procedures were performed with the Bond-Max Automated IHC Stainer (Leica Biosystems Newcastle Ltd., Melbourne, Victoria, Australia) according to the following protocol. Four-micrometer sections were cut from the paraffin blocks, deparaffinized with xylene, and pre-treated with the

Epitope Retrieval Solution 2 (EDTA buffer, pH 9.0) at 100 °C for 40 min. The sections were then incubated with the primary antibody at room temperature for 90 min. After staining with the primary antibody, sections were incubated with the polymer at room temperature for 8 min using the Bond Polymer Refine Detection Kit (Leica Biosystems, Newcastle Ltd., Newcastle Upon Tyne, UK) and then developed with 3,3′-diaminobenzidine chromogens for 10 min. Counterstaining was performed with hematoxylin.

## 3. Results

All patients included in the study showed a response to the FLOT therapy with immunotherapy. The CPS score of patients ranged from 10 to 25 percent, indicating a positive PD-L1 status (Figure 1).

We investigated the transcriptional profile of tumors and the content of autophagy-related protein, LC3B (Table 3). We expressed the mRNA level in relative units and analyzed changes in gene expression between the test sample and the reference sample. We studied the components of AKT/mTOR signaling cascade (4EBP1, AKT, c-RAF, mTOR, GSK-3β, 70S 6 kinase, PDK1, PTEN), transcriptional factors (NF-kB p65, NF-kB p50, HIF-1, HIF-2), growth factors (VEGF. VEGFR2, CAIX), PD-1, PD-L1, PD-L2, VHL, AMPK and LC3B. Additionally, we measured the protein levels of LC3B, an autophagy-related protein, in both cancerous and adjacent non-transformed tissues (represented as a percentage) (Figure 2).

**Table 3.** Molecular characteristics of tumors included in the study.

| Indicator | Tumor 1 | Tumor 2 | Tumor 3 |
|---|---|---|---|
| 4EBP1 expression, Relative Units | 2.88 | 1.97 | 0.13 |
| AKT expression, Relative Units | 1.27 | 0.50 | 1.00 |
| c-RAF expression, Relative Units | 1.56 | 0.30 | 1.00 |
| GSK-3β expression, Relative Units | 0.86 | 0.79 | 2.46 |
| 70S 6 kinase expression, Relative Units | 0.72 | 1.07 | 1.00 |
| mTOR expression, Relative Units | 0.97 | 0.54 | 1.00 |
| PDK1 expression, Relative Units | 0.54 | 0.93 | 4.00 |
| PTEN expression, Relative Units | 2.12 | 0.54 | 2.00 |
| NF-kB p65 expression, Relative Units | 2.81 | 0.74 | 1.41 |
| NF-kB p50 expression, Relative Units | 1.36 | 0.19 | 0.25 |
| VEGFR2 expression, Relative Units | 0.58 | 0.25 | 1.00 |
| VEGF expression, Relative Units | 0.18 | 0.21 | 0.03 |
| CAIX expression, Relative Units | 0.32 | 0.47 | 1.00 |
| HIF-1 expression, Relative Units | 27.21 | 0.04 | 16.00 |
| HIF-2 expression, Relative Units | 54.05 | 0.73 | 0.16 |
| VHL expression, Relative Units | 1.10 | 0.12 | 2.00 |
| PD-1 expression, Relative Units | 0.69 | 0.66 | 0.50 |
| PD-L1 expression, Relative Units | 0.31 | 0.31 | 8.00 |
| PD-L2 expression, Relative Units | 1.91 | 0.54 | 1.00 |
| AMPK expression, Relative Units | 0.00 | 0.74 | 2.00 |
| LC3B expression, Relative Units | 1.56 | 1.12 | 2.00 |
| LC3B protein, % | 111.53 | 175.00 | 231.98 |

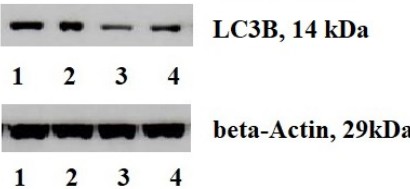

**Figure 2.** Western blotting of LC3B in cancer (1, 3) and adjacent (2, 4) tissues.

We identified the variable expression patterns in the studied indicators. The most significant changes were observed for the LC3B. Its mRNA (>1.00) and protein content (>100 percent) were high compared to the normal tissues. Although there were modifi-

cations in PD-L1 and PD-L2 mRNA level, the PD-1 expression (programmed cell death receptor) was consistently low in all cases.

## 4. Discussion

It is widely known that so-called immunity checkpoints play a crucial role in controlling the anticancer immune response. This type of immunotherapy has been successfully used to treat various types of cancers including breast, lymphoma, and colorectal cancer. Immune checkpoint blockade (ICB) has become the standard of care for several metastatic epithelial cancers and has significantly improved the life expectancy of many patients [11].

PD-1/PD-L1 pathway blockade works as an effective and practical therapy for gastric cancer immunotherapy [11,12]. PD-1 and PD-L1 co-expression predicts a favorable prognosis in gastric cancer [12]. It is known that PD-L1 expression is more prevalent in men with papillary unclassified HER2/neu+ EBV+ proximal gastric cancer and PIK3CA mutation [13], associated with an MSI-High status [14,15].

A phase III Asian ATTRACTION 04 randomized study demonstrated an increase in the overall survival rate in patients treated with immune checkpoint inhibitors [13]. The KEYNOTE-059 clinical trial (Cohort 1; Phase II) showed a higher objective response rate in patients with positive PD-L1 status who received pembrolizumab (PD-1 inhibitor) in combination with capecitabine and cisplatin compared to those with a negative status [15]. The phase III KEYNOTE-059 study included a third non-PD-L1+/− cohort of patients, and it further supported the higher response rate in PD-L1-positive patients who underwent chemotherapy with anti-PD-1 drugs [16].

The neoadjuvant therapy with immune check-point blockage was found to be highly effective, which is associated with changes in dominant tumor subclones and immune microenvironments [17]. Consequently, cellular biological processes are responsible for the anticancer therapy effect. The most powerful one is autophagy, which determines the aggressive biological features of cancer [6,7]. Our previous research showed an increase in LC3B content with reduced response to FLOT chemotherapy [11]. The studied molecular indicators also revealed autophagy activation in gastric cancers with positive PD-L1 status. It is a sign of unfavorable prognosis in cancers. Additionally, a decrease in PD-1 expression in tumors was revealed. It was found to be essential for antitumor response [18]. Therefore, the combination of autophagy activation and a lack of antitumor immunity may be significant molecular features in PD-L1-positive gastric cancers.

Recently, it has been suggested that tumor molecular profiling could improve the treatment of patients with gastric cancers [19]. Evidently, genomic alterations in gastric cancers affect the therapy response and patient survival [20]. Nevertheless, this classification was not able to develop a prognostic stratification system for gastric cancer. NGS technology is unacceptable for routine molecular diagnostics. It could not reveal any prognostic indicators associated with cancer biology.

Currently, there is a significant focus on finding prognostic indicators for cancer based on oncogenic processes. Implementing these indicators in clinical practice will help practitioners to choose a personalized therapy for patients. Despite the worldwide use of immune checkpoint inhibitors in the treatment of gastric cancer, there are biological processes that hinder their effectiveness. Therefore, improving the efficacy and response rates of immunotherapy has become a challenge. The most promising indicators are the autophagy-related proteins and the level of PD-1 receptors. Our analysis of clinical data highlights the biological features of effective anticancer therapy, but further investigation is needed.

## 5. Conclusions

Thus, we have highlighted the clinical cases that demonstrate the high efficacy of combining FLOT chemotherapy with immune checkpoint inhibitors in gastric cancer patients who have a PD-L1-positive status. These tumors exhibit autophagy activation,

which leads to changes in immunity. However, further investigation is necessary to identify the molecular and biological indicators associated with effective anticancer treatment.

**Author Contributions:** Conceptualization, L.V.S.; methodology, A.V.A.; formal analysis, O.V.B.; investigation, L.V.S.; resources, S.G.A.; data curation, A.M.V.; writing—original draft preparation, L.V.S.; writing—review and editing, L.V.S. All authors have read and agreed to the published version of the manuscript.

**Funding:** This work was financially supported by the Government research assignment for ISPMS SB RAS, project FWRW-2022-0002.

**Institutional Review Board Statement:** The study was conducted in accordance with the guidelines of the Declaration of Helsinki and approved by the Institutional Review Board of Cancer Research Institute, Tomsk National Research Medical Center (protocol code 4; dated 16 November 2018).

**Informed Consent Statement:** Written informed consent has been obtained from the patient(s) to publish this paper.

**Data Availability Statement:** The data that support the findings of this study are openly available in open journals at http://doi.org/10.3390/cimb44070190; http://doi.org/10.2174/1381612826666201120155120 accessed on 2 September 2023.

**Conflicts of Interest:** The authors declare no conflict of interest.

## Appendix A

Figure 1 shows high-grade adenocarcinoma with cricoid cells. PD-L1 tumor status positive (CPS-60). Figure 2 shows western Blot image of LC3 content in gastric cancers.

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
