# Peer review of "Safety and Efficacy of Neoadjuvant Chemoimmunotherapy in Gastric Cancer Patients with a PD-L1 Positive Status: A Case Report"

_cimb, doi:10.3390/cimb45090481_

Round 1

Reviewer 1 Report

Unfortunately the submitted manuscript is not on par with the journal's high standars. English language needs heavy editing, the inclusion criteria of the subjects are inadequately explained, some content that should be present in Results section, is present in Materials and Methods. Discussion is poor due to the fact that the benefits/limitations of this case series report, are not clearly explained.

Extensive editing of English language is required through the entire text. Many syntax and grammar errors are present in the text.

Author Response

The team of authors thanks for the thorough review of the article. All comments were taken into account. An abstract was rewritten, to which there were many comments. Changes were also made to the materials and methods of the study. The clinical material was supplemented with molecular studies. The discussion section has been moved from the results to the discussion. Discussion section. as well as the conclusion were rewritten using new data. English correction was made.

Reviewer 2 Report

Title of the manuscript: Safety and efficacy neoadjuvant chemoimmunotherapy in patient with PD-L1 positive status gastric cancer: A case series. I really appreciate the work and the endeavor of the authors and based on these clinical cases and the previous works I think the authors are specialists in this topic of oncology. However, I do not feel the strengths of the present work, I have found a great number of methodological/grammatical mistakes, so I do not think that the manuscript in the present form is suitable for publication.

My concerns/notifications are the following:

1, The reviewer is not a native English speaker, however the English of the manuscript is found to be too weak, sometimes there are no nouns (e.g., lines 14-15) or no verbs (e.g., line 16) in some sentences. It is absolutely recommended to overview the material by an experienced English language lecturer once more and before the next submission.

2, Nor the Title and neither The Abstract is informative, and there is no description about the significance of the complete pathological regressions in the clinical cases. Moreover, the reader cannot find any special interest in the abstract, that would be attractive for an expert.

3, In the Material and Method section there could be found a great number of ambiguous descriptions and/or misinterpretation, e.g., body and subtotal localization (??) (see Table 1.), median and later (in separate paragraph) average ages and ranges in case of 3 patients (!!),  first the authors described 2 partial regression, later on 2 partial regression and 1 stabilization (based on CT follow-up), however, the pathological finding was total regression in all cases (and there is no explanation for this phenomenon). It is a little bit strange to determine the percentiles in case of 3 patients, and there is no talk about the postoperative treatments/events. Summarizing these findings, the reader feels that this this section is a little bit confusing.

4, The Results section is a kind of a pre-Discussion with some general statement and the description of one clinical study. Over the interesting medical data of this section, the most extraordinary finding that one paragraph is repeated in three (!!) times (see lines 142-148, 148-154, 154-159).

5, The Discussion section is only talking about the results of immunotherapy in inoperable/locally advanced/metastatic cases/settings, and there is no relevant data about the results of neoadjuvant immunotherapy, however there could be find more than 30 running clinical trials in this indication (see clinicaltrials.gov).

Conclusively, over some real value and curiosity of the manuscript, I do not find enough clinically relevant new data in it considering molecular medicine, and I do not think that the work is worthwhile for publication in the present form.  

Author Response

The team of authors thanks for the thorough review of the article.

The aim of the paper was changed. The content of the paper was also modified to clarify the molecular-based approach in gastric cancers treatment.

  1. The proofreading was made. All corrections were done that improve the quality of the paper..
  2. The abstract was rewritten. Its sections were modified and corrected
  3. The material and methods section was modified the definitions were changed. We hope that all corrections improve the paper.
  4. Results section was modified. The discussion was transferred to the discussion part.

5, The Discussion section was modified and neoadjuvant chemotherapy was added to the paper

  1. The molecular based explanation of the good effect of the anti-PD-L1 therapy was found. It is obtained the need in further investigation.

Reviewer 3 Report

This case report was concerned with the clinical responses of three patients with gastric cancer to the concurrent treatment with FLOT and 400 mg of pembrolumab for 6 weeks. The results indicated that the treatment induced adverse effects were tolerable in all three patients. Upon the completion of the treatment cycle, two patients showed partial response and one achieved stable disease. 

Major Comments:

1) Contents in Line 93 - Line 105 should be moved to the Results section.

2) Individual patients' IHC results shown in Figure 1 need further elaboration. If a scoring system is used, please clearly indicate the PD-L1 expression score in each patient along with their response to the treatment. 

3) The results section read more like discussions and presented little data from the authors' own study. 

4)  Line 146 - 150 need to be rewritten.  When the authors stated that "The authors concluded that high PD-L1/PD-1 expression was significantly associated with better overall survival", they need a statistical analysis to support their statement. Since there were only three patients recruited in this study, the conducting statistical analysis on the results would be difficult. Therefore, it is inappropriate to claim that there would be a significant effect. 

Extensive editing of English language required

Author Response

The team of authors thanks for the thorough review of the article.

The aim of the paper was changed. The content of the paper was also modified to clarify the molecular-based approach in gastric cancers treatment.

  1. The proofreading was made. All corrections were done that improve the quality of the paper..
  2. The abstract was rewritten. Its sections were modified and corrected
  3. The material and methods section was modified the definitions were changed. We hope that all corrections improve the paper.
  4. Results section was modified. The discussion was transferred to the discussion part.

Reviewer 4 Report

The case report entitled “Safety and efficacy neoadjuvant chemoimmunotherapy in patient with PD-L1 positive status gastric cancer: A case series” is an interesting report with promising findings. The authors have reported from 3 patients that neoadjuvant immunotherapy shows favorable pathological responses in patients with significant PD-L1 positive status of the gastric tumors.

Limitations: Only 3 patients. Hence any statistical significance is lacking.

The manuscript has serious writing flaws that need to be addressed by an expert in the field as well as the English language.

Comments:

1.       The title should be “Safety and efficacy of neoadjuvant chemoimmunotherapy in a patient with PD-L1 positive status gastric cancer: A case report”.

2.       The sentence “Three patients managed at our institution received neoadjuvant chemotherapy FLOT and immunotherapy and surgery for advanced, resectable.” in the abstract looks incomplete. Correct it.

3.       In the abstract “Next-generation sequencing (NGS), immunohistochemical (IHC) staining” should be written in sentence form.

4.       Break this sentence “The effectiveness of early diagnosis remains low, in 2019 the share of common processes (GC II-III stages) accounted for 44.9% of diagnosed cases of the disease, advanced cancer was diagnosed in 39.9% of patients, which largely determines unsatisfactory long-term results treatment [3, 4]” into smaller sentences.

5.       Elaborate RJ in the sentence “The Cancer Genome Atlas Research Network proposes the following molecular classification of RJ”

6.       The authors have used abbreviations before first mentioning the complete forms in many places. Rectify it.

7.       Write in 2 sentences “Sequencing (NGS) was performed to detect the MSI status Immunohistochemical (IHC) staining or in situ hybridization (ISH) analysis was used to determine the expression status of PD-L1, HER-2.”

8.       Line 75, write one female and not females.

9.       Add the sex of the patients in Table 1.

10.   Line 104, the correct spelling of investigation.

11.   Most of the parts that authors have written in the material and methods sections should go in the results section. The figure should be the part of results and not the materials and methods section.

12.   In the results section, authors have repeatedly used “The authors concluded that high PD-L1 expression was significantly associated with better overall survival. 4% of primary tumors and 73.3% of metastases. PD-1 expression was found in tumor-infiltrating lymphocytes in 53.8% of primary gastric cancer tumors and in 73.3% of liver metastases. Of note, PD-L1 expression is significantly more prevalent in men with papillary unclassified HER2/neu+ EBV+ proximal gastric cancer with PIK3CA mutation and increased levels of microsatellite instability (MSI).” Avoid repetition and proofread the manuscript properly before submitting again.

13.   Introduction and discussion section should be improved with adequate references. 

Extensive editing of English is required.

Author Response

The team of authors thanks for the thorough review of the article.

  1. The title should be “Safety and efficacy of neoadjuvant chemoimmunotherapy in a patient with PD-L1 positive status gastric cancer: A case report”.

The title was changed

  1. The sentence “Three patients managed at our institution received neoadjuvant chemotherapy FLOT and immunotherapy and surgery for advanced, resectable.” in the abstract looks incomplete. Correct it.

Done. The sentence was changed.

  1. In the abstract “Next-generation sequencing (NGS), immunohistochemical (IHC) staining” should be written in sentence form.

The abstract was rewritten

  1. Break this sentence “The effectiveness of early diagnosis remains low, in 2019 the share of common processes (GC II-III stages) accounted for 44.9% of diagnosed cases of the disease, advanced cancer was diagnosed in 39.9% of patients, which largely determines unsatisfactory long-term results treatment [3, 4]” into smaller sentences.

The sentence was rewritten

  1. Elaborate RJ in the sentence “The Cancer Genome Atlas Research Network proposes the following molecular classification of RJ”

All corrections were made

  1. The authors have used abbreviations before first mentioning the complete forms in many places. Rectify it.

Abbreviations were included in the paper

  1. Write in 2 sentences “Sequencing (NGS) was performed to detect the MSI status Immunohistochemical (IHC) staining or in situ hybridization (ISH) analysis was used to determine the expression status of PD-L1, HER-2.”

The sentence was rewritten.

  1. Line 75, write one female and not females.

The sentence was rewritten

  1. Add the sex of the patients in Table 1.

The table was modified

  1. Line 104, the correct spelling of investigation.

Spelling was corrected

  1. Most of the parts that authors have written in the material and methods sections should go in the results section. The figure should be the part of results and not the materials and methods section.

The results section was modified

  1. In the results section, authors have repeatedly used “The authors concluded that high PD-L1 expression was significantly associated with better overall survival. 4% of primary tumors and 73.3% of metastases. PD-1 expression was found in tumor-infiltrating lymphocytes in 53.8% of primary gastric cancer tumors and in 73.3% of liver metastases. Of note, PD-L1 expression is significantly more prevalent in men with papillary unclassified HER2/neu+ EBV+ proximal gastric cancer with PIK3CA mutation and increased levels of microsatellite instability (MSI).” Avoid repetition and proofread the manuscript properly before submitting again.

The section was modified

  1. Introduction and discussion section should be improved with adequate references. 

Adequate references were added to the introduction

Round 2

Reviewer 1 Report

English language (syntax and grammar) is incomprehensible at many points throught the manuscript, and does not fit the quality of this journal. Please, edit the text with the help of a native English language speaker.

English language (syntax and grammar) is incomprehensible at many points throught the manuscript, and does not fit the quality of this journal.

Author Response

The english proofreading was made.

Reviewer 2 Report

The second version of the manuscript has really improved; however, I still have several existing and significant notifications/ recommendations/ concerns.

-I suggest a novel English language overview of the material.

-I have not found the basic message of the work nor in the title, neither in the abstract.

-Abstract: The first and the second sentences of the abstract are totally unclear. Moreover, the reader cannot realize if the message is mainly clinical or mainly molecular pathology based. Maybe it would be emphasized the significance of autophagy in the immune system work.  

-If the frequencies of Stage II-III tumors are 44,9 and 39,9%, and early Stage I neoplasms are also detected, the percentage of de novo metastatic Stage IV tumors is surprisingly very low. Please control the data.

-Table1: The ages of the patients absolutely differ in the Text and in the Table. Please correct it! I do not know the “subtotal” localization. ypT0N0 (without M0) is a postoperative pathology category after primary systemic therapy. Please clarify/correct these data in the Table.

-What is the explanation/in the background considering the difference between the radiological and pathological response after combined immune-chemotherapy?

-The paragraph between lines 101 and 125, and Table 2 are not enough informative and unclear for a clinician reader. Please format them once more.

-I suggest a more detailed debate in the Discussion section about the molecular subtypes/ molecular alterations with different immune/clinical reactions.

Author Response

Response to Reviewer 1 Comments

Point 1:

-I suggest a novel English language overview of the material.

Response 1

The proofreading was made

Point 2:

-I have not found the basic message of the work nor in the title, neither in the abstract.

Response 2

We found the high effectiveness of FLOT chemotherapy with “immune check-points” in gastric cancers patients with PD-L1 positive status. Autophagy was involved in activating tumor immunity. Further investigation needs to clarify the mechanism of effective anticancer treatment.

Point 3:

-Abstract: The first and the second sentences of the abstract are totally unclear. Moreover, the reader cannot realize if the message is mainly clinical or mainly molecular pathology based. Maybe it would be emphasized the significance of autophagy in the immune system work.  

Response 3

The landscape of gastric cancer treatment has changed owing to the widespread use of immune checkpoint inhibitors. Autophagy regulating immune system is a potential trigger of immunity in tumors.

Point 4 :

-If the frequencies of Stage II-III tumors are 44,9 and 39,9%, and early Stage I neoplasms are also detected, the percentage of de novo metastatic Stage IV tumors is surprisingly very low. Please control the data.

Response 4

The percent of cancers with II-III stages ranged from 39.9% to 44.9% in gastric cancer patients

Point 5 :

-Table1: The ages of the patients absolutely differ in the Text and in the Table. Please correct it! I do not know the “subtotal” localization. ypT0N0 (without M0) is a postoperative pathology category after primary systemic therapy. Please clarify/correct these data in the Table.

Response 5

The data in the table and in the paper was corrected

Point 6 :

-What is the explanation/in the background considering the difference between the radiological and pathological response after combined immune-chemotherapy?

Response 6

There is a difference in pathological response between the radiological therapy and combinedimmune-chemotherapy.

Point 7 :

-The paragraph between lines 101 and 125, and Table 2 are not enough informative and unclear for a clinician reader. Please format them once more.

Response 7

The mRNA level was presented in relative units and was analyzed changes in gene expression in a given sample relative to another reference sample. The components of AKT/mTOR signaling cascade (4EBP1, AKT, c-RAF, mTOR, GSK-3β, 70S 6 kinase, PDK1, PTEN), transcriptional factors (NF-kB p65, NF-kB p50, HIF-1, HIF-2), growth factors (VEGF. VEGFR2, CAIX), VHL, AMPK and LC3B were investigated. The protein level of autophagy-related protein LC3B was also measured in cancers and adjucent non-transformed tissues.

Point 8 :

-I suggest a more detailed debate in the Discussion section about the molecular subtypes/ molecular alterations with different immune/clinical reactions.

Response 8

Recently, it was declared that tumor molecular profiling could improve the treatment of patients with gastric cancers [19]. Evidently, genomic alterations in gastric cancers affect the therapy response and patient’s survival [20]. Nevertheless, this classification failed to construct a prognostic stratification system for gastric cancer. NGS technology is unacceptable for the routine molecular diagnostics.

Reviewer 3 Report

The authors have addressed all of my questions. 

Author Response

Authors modified the paper and made any corrections.

Reviewer 4 Report

The case report entitled “Safety and efficacy neoadjuvant chemoimmunotherapy in patient with PD-L1 positive status gastric cancer: A case report” has been extensively revised by the authors.

The authors have missed certain suggestions which are mentioned below. Also, a couple of other corrections are needed.

1.       The title should be “Safety and efficacy of neoadjuvant chemoimmunotherapy in a patient with PD-L1 positive status gastric cancer: A case report”. Authors have missed “of and a”.

2.       In the abstract, correct PD-L1 mentioned twice.

3.       Add the sex of the patients in Table 1. I didn’t see this modification in the table.

4.       Authors should move Figure 1 to the results section and describe histopathology results in the results section.

5.       The list of primers can be given in tabular form.

6.       There are still multiple spelling mistakes like in the sentence in the results section “The CPS score of patients raged betwenn the 10 and 25 percent” and “The most signioficant changes were obtained for the LC3B.”. Ranged, between, and significant. Carefully go through the manuscript to correct these mistakes and some of the grammatical errors.

7.       Authors have mentioned that “The LC3B content was measured by Wester Blotting analysis”. They should show it in the results section.

Moderate editing is required.

Author Response

Response to Reviewer 4 Comments

 Point 1:

  1. The title should be “Safety and efficacy ofneoadjuvant chemoimmunotherapy in a patient with PD-L1 positive status gastric cancer: A case report”. Authors have missed “of and a”.

 Response 1

The title was changed

Safety and efficacy of a neoadjuvant chemoimmunotherapy in patients with PD-L1 positive status gastric cancer: A case report

 Point 2:

  1. In the abstract, correct PD-L1 mentioned twice.

  Response 2

Corrections were made

  Point 3:

  1. Add the sex of the patients in Table 1. I didn’t see this modification in the table.

   Response 2

Corrections were made

 Point 4:

  1. Authors should move Figure 1 to the results section and describe histopathology results in the results section.

Response 4

The figure was transferred to the Results section.

Point 5:

  1. The list of primers can be given in tabular form.

 Response 5

The table with primers sequences was added.

Point 6:

  1. There are still multiple spelling mistakes like in the sentence in the results section “The CPS score of patients raged betwenn the 10 and 25 percent” and “The most signioficant changes were obtained for the LC3B.”. Ranged, between, and significant. Carefully go through the manuscript to correct these mistakes and some of the grammatical errors.

  Response 6

Corrections were made

Point 7:

  1. Authors have mentioned that “The LC3B content was measured by Wester Blotting analysis”. They should show it in the results section.

Response 7

The figure with the results is incorporated to the paper

Round 3

Reviewer 2 Report

The quality of the material has really improved. I have only some remnant notifications.

1, Equalize the age of the "median" patient (53 vs. 58 years) see text (line 73. and table 1.)

2, What is subtotal localisation? (Table 1.)

3, There are different check-point inhibitors in the clinical routine (like CTLA4 inhibitors, PD1 inhibitors, PDL1 inhibitors etc.)  see e.g. line 144.

4, Pembrolizumab is a PD1 inhibitor (line 157.)

- I suggest less letters in Table 2.

- It would be interesting to learn more about the role of immune autophagy and the LC3B marker.  

Author Response

An abstract was rewritten, to which there were many comments. Changes were also made to the materials and methods of the study.

Comment 1

1, Equalize the age of the "median" patient (53 vs. 58 years) see text (line 73. and table 1.)

Response 1

Median was calculated and changed in the paper.

Comment 2

2, What is subtotal localisation? (Table 1.)

Response2

The localization of the tumor was clarified. (body and pyloric antrum)

Comment 3

3, There are different check-point inhibitors in the clinical routine (like CTLA4 inhibitors, PD1 inhibitors, PDL1 inhibitors etc.)  see e.g. line 144.

Response3

Checkpoint inhibitors are a type of immunotherapy. They are a treatment for cancers such as melanoma skin cancer and lung cancer. These drugs block different checkpoint proteins. You might also hear them named after these checkpoint proteins – for example, CTLA-4 inhibitors, PD-1 inhibitors and PD-L1 inhibitors. We used the PD-1 inhibitors

Comment 4

4, Pembrolizumab is a PD1 inhibitor (line 157.)

- I suggest less letters in Table 2.

- It would be interesting to learn more about the role of immune autophagy and the LC3B marker.  

Response 4

Pemrolizumab is PD-1 inhibitor. We changed all mistakes.

Autophagy research also includes its role in immunity . The specificity of the immune response is provided by proteins and cells circulating the body. Different mechanisms contribute to the immune response and can be put into two divisions, adaptive immunity and non-adaptive immunity. . LC3 is an important autophagy 8 8 marker in that it can be used to follow and analyze the selection of protein and cellular structures into the autophagy pathway and to the formation of autophagosomes within the cells. The formation of immunity can use this mechanism.

Reviewer 4 Report

The case report entitled “Safety and efficacy of a neoadjuvant chemoimmunotherapy in a patient with PD-L1 positive status gastric cancer: A case report” was revised but still, some changes are needed.

In Figure 2, there seems to be unequal loading of the proteins. The bands of b-actin in lanes 2 and 4 are very faint compared to other lanes. If an equal amount of the protein is loaded in all the lanes, then the bands of b-actin (as a loading control) should look equally intense. Authors need to repeat the experiment to show proper blots, quantify the results (normalize with the loading control), and show it as a graph next to the western blot figure.

Moderate editing is still required.

Author Response

Response to Reviewer 1

The team of Authors revised again the paper and the figures of the Western Blotting analysis

Point 1

In Figure 2, there seems to be unequal loading of the proteins. The bands of b-actin in lanes 2 and 4 are very faint compared to other lanes. If an equal amount of the protein is loaded in all the lanes, then the bands of b-actin (as a loading control) should look equally intense. Authors need to repeat the experiment to show proper blots, quantify the results (normalize with the loading control), and show it as a graph next to the western blot figure.

Response 1

The figure 2 was modified according to the Reviewer’s comments